# Establishment of a Palliative Care Consultation Service (PCCS) in an Acute Hospital Setting

**DOI:** 10.3390/ijerph17144977

**Published:** 2020-07-10

**Authors:** Peter Tom Engel, Tharshika Thavayogarajah, Dennis Görlich, Philipp Lenz

**Affiliations:** 1Department of Palliative Care, University of Muenster, 48149 Muenster, Germany; peter.tom.engel@gmx.de; 2Department of Hematology, Oncology, Hemostaseology, and Stem Cell Transplantation, Faculty of Medicine, RWTH Aachen University, 52074 Aachen, Germany; tharshika.thavayogarajah@rwth-aachen.de; 3Institute of Biostatistics and Clinical Research, University of Muenster, 48149 Muenster, Germany; Dennis.Goerlich@ukmuenster.de

**Keywords:** early integration, palliative care, palliative care consultation service

## Abstract

*Background and study aims*: Acute health service requires focused palliative care (PC). This study was performed to provide guidance for the establishment of a palliative care consultation service (PCCS). *Patients and methods*: This study was conceived as a retrospective single-center study for observing, analyzing and evaluating the initial setup of a PCCS from 1 May 2015 to 31 May 2018. Patients from Muenster University Hospital with advanced life-limiting diseases, identified to require PC, were included. *Results*: PCCS was requested from various departments, for between 20 and 80 patients per month, corresponding to a total of 2359 for the study period. Requests were highest in internal medicine (27.3%), gynecology (18.1%) and radiotherapy (17.6%). Time to referral was significantly shorter in departments with special PCCS ward rounds (6 ± 9 vs. 12 ± 22 days, *p* < 0.001). The most frequently reported symptoms were fatigue, pain and loss of appetite. Pain was frequently localized in the stomach (20.4%), back (17.1%), or in the head and neck area (14.9%). After the first PCCS consultation, 254 patients (90%) reported sufficient pain relief after 48 h. An introduction/modification of painkiller medication, which was recommended for 142 inpatients, was implemented in 57.0% of cases by the respective departments. Overall, the direct realization of PCCS recommendations reached only 50% on average. *Conclusions*: Besides an analysis of the ability to address the symptoms of the referred patients by the PCCS, this study highlights the importance of the interaction between PCCS and other departments. It further elucidates the role and possibilities of this service both in regular ward rounds and individual staff contacts.

## 1. Introduction

Palliative care (PC) supports patients with life-limiting disease and their relatives by improving quality of life. “Pain and other problems, physical, psychosocial, and spiritual”, are parts of the assessment and treatment [1]. The most frequent symptoms of PC patients are pain, lack of energy, fatigue, weakness and appetite loss [2].

Strikingly, Becker et al. reported that 9.1% of all patients older than 65 years would require PC [3]. This indicates a considerable demand for PC. Although PC in Germany has improved continuously in recent years, an increase in the availability of PC services for patients is still required: there are only 15 opportunities for PC service, plus 17 hospice beds, per million inhabitants. About 17–18% of incurable tumor patients need PC, but the current capacities allow only 3–4% to receive adequate care. Moreover, there is wide variation in the distribution of PC and hospice beds in Germany [4].

Several different estimates have been made regarding the demand for PC beds in Europe [5]. Considering cancer patients only, around 50 beds per one million inhabitants are needed [5]. Notably, non-cancer patients, and the increasing prevalence of chronic diseases due to demographic changes, are not considered in these data. Considering these two groups of patients, the European Association for Palliative Care (EAPC) currently estimates an actual necessity of 80–100 PC and hospice beds per one million inhabitants [5].

A palliative care consultation service (PCCS) supports physicians and nurses in both in- and outpatient departments by providing medical recommendations for the respective patients. It must be noted, though, that it is incumbent upon the physicians and nurses in charge to make the final decision about a patient’s treatment [6].

The advantage of a PCCS is the ability to consult a large number of patients within a short period in the hospital. Particularly, an outpatient setting allows for easy and early access to patients [7].

Use of a PCCS also improves pain and symptom control, while reducing anxiety and hospital admissions [8]. Further, communication with the PCCS team enhances patient satisfaction regarding hospital care [9]. Moreover, contact with the PCCS improves the quality of dying (QOD) as reported by relatives [10]. In this context, QOD must be a multidimensional concept, e.g., including physical, psychological, social, and spiritual or existential experiences [11].

Thus, a PCCS is needed for every hospital in addition to a PC ward, and every hospital should have access to a PCCS if required [6]. Furthermore, a hospital with 250 beds should have at least one PCCS team [5].

Cancer patients facing progressed stages of their disease should be consulted by a member of the PCCS [12]. PCCS is most efficient when having early contact with patients who are diagnosed with life-limiting diseases, and for those patients who request PC due to their medical circumstances [12,13]. Early contact with the PCCS may facilitate an outcome associated with a high quality of life, such as less aggressive medical treatments and fewer inpatient admissions in the last months of life, compared to standard care [14].

Palliative care also has a wider dimension: PC treatment involves patients, nurses, doctors and relatives who all have different perceptions concerning their “own truth” of the situation [15]. In this context, it is necessary to recognize the most important needs of patients and family members in the context of end-of-life care, as found in a study: “*trust in the treating physician, avoidance of unwanted life support, effective communication, continuity of care and life completion*” [16]. 

This study aimed to provide guidance regarding how to structure, establish, and set up a PCCS in an acute hospital setting. We investigated the type and severity of symptoms presented to the PCCS, and analyzed the overall characteristics of patients referred to our PCCS, as well as the realization of our recommendations by the referring department. Furthermore, we hypothesized the existence of distinct patterns of symptoms in different patient cohorts, so that differences in the presence of symptoms of referred patients from different departments could be comparatively analyzed. In addition, we hypothesized that departments with regular PCCS ward rounds would refer patients faster to the PCCS as compared to departments without established PCCS ward rounds.

More specifically, each medical department could involve our PCCS if there was a need for PC treatment. Requests were answered within 24 h. Our PCCS team focused its attention on the anamnesis of the existence and severity of symptoms and conditions of the referred patients, and subsequently made recommendations aiming at the control or relief of these symptoms; however, the final decision regarding further medical treatment was made by the referring department and, thus, the PCCS took on an advisory role.

## 2. Patients and Methods

### 2.1. Study Design

This study was designed as a retrospective, single-center study at the Muenster University Hospital, where a PCCS was established in May 2015. For the following study, we analyzed the first years of building up a PCCS from 1 May 2015 to 31 May 2018. In the subgroup analysis, we include data from the first 13 months after establishing the PCCS.

### 2.2. Patients

Patients with advanced life-limiting and progressive disease were referred to the PCCS by ward physicians when, together with their team, a need for PC was identified. A specialized palliative care (SPC) physician or nurse performed the initial PC assessment, including a detailed symptom burden assessment (symptom distress scale; McCorkle and Young) [17].

Focusing on the individual needs, the patients were processed according to the algorithm shown in Figure 1 [18]. 

All data were electronically available using the hospital information system Orbis-OpenMed^®^ (Agfa Healthcare, Mortsel, Belgium).

The Eastern Cooperative Oncology Group (ECOG) score was used to assess patient performance regarding activities of daily living [19]. ECOG consists of six grades from 0 to 5 describes an unrestricted activity; *1* activity restriction, but the ability to work; *2* the inability to work, but the ability for self-supply; *3* a restricted self-supply with 50% of the time confined to bed or chair; *4* complete dependence on care; *5* means death [19]. 

Symptom burden was assessed in 6 categories: stable (asymptomatic), slightly reduced, reduced, greatly reduced, unbearable, and no data. These were summarized and color-coded in a heatmap in the following manner: green (stable), yellow (slightly reduced and reduced), red (greatly reduced and unbearable) and white (no data). 

### 2.3. Team Structure

The theme introduced in PC by Dame Cicely Saunders, “Low technology, high personality”, describes the importance of the team members in PC. Our core team consists of two physicians, two nurses, two physiotherapists, a half-time weekly psycho-oncologist, a half-time weekly social worker, and a music therapist. Dieticians, pain therapists and chaplains support the core team. 

### 2.4. Statistics

Continuous variables are summarized by the mean and standard deviation. Categorical variables are presented as absolute and relative frequencies. 

Time to referral was defined as the time from hospital admission to first contact with the PCCS. To test if the time to referral differed between departments with vs. without a PCCS ward round, the Mann–Whitney U test was used. A two-sided *p*-value of < 0.05 was considered to indicate statistical significance.

Symptom burden presented at first PCCS consultation was visualized on a scale of stable (1; green), slightly reduced (2; yellow), reduced (3; yellow), greatly reduced (4; red) and unbearable (5; red), and presented as a symptom heatmap sorted by patient and symptoms.

A comparison of the realization of recommendations between males and females for the respective symptom was analyzed using the χ^2^ test. 

The statistical analysis of the data was performed using the SPSS software (IBM Corp. Released 2012. IBM SPSS Statistics for Windows, Version 21.0. IBM Corp., Armonk, NY, USA) and SAS software (Version 9.4, SAS Institute Inc., Cary, NC, USA). The symptom heatmap was generated using the ggplot2 package [20] within the statistical software R (Version 3.2.0 for Windows, http://www.R-project.org/).

## 3. Results

### 3.1. Procedural Characteristics

We analyzed a total of 2359 patients in the period from May 1, 2015 to May 31, 2018. Of all patients, 1946 (82.5%) were inpatients and 413 (17.5%) outpatients. All were seen by the PCCS, and within 12 months, the number of patients increased from 20 to 60–80 per month at the end of the study period. The number of patients per year is shown in Figure 1B. 

### 3.2. Time to Referral—Inpatients 

We defined the time from hospital admission to first contact with the PCCS as the time to referral. The median time to referral in the observation period was 10 ± 19 days, with a minimum of 0 and a maximum of 332 days. In departments with special PCCS ward rounds, the median time to referral was significantly shorter compared to the departments without ward rounds (6 ± 9 vs. 12 ± 22 days, *p* < 0.001).

### 3.3. Referring Departments—Inpatients 

All departments with inpatient care (*n* = 18) requested PCCS support for their treatment. The PCCS established initial regular weekly ward rounds at the department of radiotherapy and gynecology; later, rounds were also included for the department of hematology/oncology. 

As a result, the following three departments had the highest requests for PCCS for their patients: internal medicine (27.3%), gynecology (18.1%) and radiotherapy (17.6%). The data for the other departments are presented in Figure 1C. 

### 3.4. Subgroup Analysis

The following subgroup analysis displays the data for 578 patients, covering the period from 1 May 2015 to 31 May 2016; however, the parameter “pain relief within 48 h after PCCS consultation” was introduced later, and includes the following period: 1 July 2016 to 31 May 2018 (*n* = 1635). 

### 3.5. Patient Characteristics

From a total of 578 patients, 490 were inpatients and 88 outpatients. The demographic data of our inpatients, including their advanced healthcare directive (AHCD) and healthcare proxy, can be found in Table 1.

Further, 88% of our patients had cancer and 12% had other diseases (429 patients with cancer vs. 61 with other diseases). The fields with the most common diagnoses in our subgroup analysis were gastrointestinal (12.9%), gynecological (8.4%), head and neck (8.3%) and genitourinary cancer (7.4%). 

Of the solid cancer patients (excluding brain tumors, *n* = 350), metastatic disease was observed in 80.3%, and peritoneal carcinosis in 11.7%. Interestingly, the majority of the evaluated patients were found to be ECOG 3, which means a restricted self-supply with 50% of the time confined to bed or chair (Figure 2B). 

### 3.6. Symptom Burden

We generated a heatmap to create a first impression of our patient’s symptom burden (Figure 2C). The most commonly presented symptoms were fatigue (82.7%), depression (68.6%) and loss of appetite (64.7%). Moreover, some patients reported constipation (62.7%), pain (56.7%; for detail, see below) and insomnia (54.9%). 

Less frequently, symptoms like dyspnea (43.2%), nausea (30.4%), edemata (28.7%), vomiting (20.8%) and wounds (19.2%) were presented. Most of our patients could not be assessed regarding neurologic symptoms (56.9%) or bleeding risk (56.2%). 

### 3.7. Symptom Burden in Different Departments

Figure 3 shows that some symptoms are more frequent in specific departments. For example, in hematology and oncology, fatigue (96.6%), depression (82.8%) and loss of appetite (73.6%) were the three most frequent symptoms, similar to the symptom burden of all our patients; however, dyspnea (71.3%), constipation (64.4%), insomnia (62.1%) and edemata (57.5%) were more frequent in hematology and oncology patients than in the group of all analyzed patients. 

### 3.8. Pain

Moreover, we used the visual analog scale (VAS), from 0 (no pain) to 10 (strongest imaginable pain), to assess patient pain levels. As a result, 278 (56.7%) of our patients suffered pain, with 28.0% of them in low (1–3), 42.0% in middle (4–7), and 23.9% in high (8–10) pain (no data in 6.1%; mean 5.1 ± 2.7; median 5.0). 

Of our patients who suffered pain (*n* = 278), the pain was mostly located in the stomach (20.4%), back (17.1%), or in the head and neck area (14.9%) (Figure 2A). 

#### 3.8.1. Premedication and Changes Due to PC Consultation

##### Pain

In our analysis, 414 (84.5%) of our patients took one or more painkillers, and of these patients, 102 (24.6%) were in step I (non-opioid) of the WHO pain ladder, 26 (6.3%) were in step II (weak opioid ± non-opioid), and 286 (69.1%) were in step III (strong opioid ± non-opioid). 

For 142 inpatients (28.9% of all inpatients), the PCCS recommended a drug conversion for painkillers, and for 328 patients (66.9%), no change was recommended. Further, for 81 patients (57.0%), the PCCS recommendations were performed directly (within 48 h of recommendation). When analyzed according to sex, females (59.1%) experienced slightly more frequent, though non-significant, receipt of recommended pain care compared to males (55.3%).

There were four possibilities for optimizing the pain medication: (1) the pain medication could be increased by taking a step in the WHO pain ladder (20 patients, 25% of the patients with performed recommendations); (2) the opioid could be changed (15 patients, 18.8%); (3) a WHO pain ladder step could be completed, such as through adding a non-steroidal anti-inflammatory drug (NSAID) to the regimen (14 patients, 17.5%); or (4) the dose could be escalated (13 patients, 16.3%).

If a new opioid was added or there was a conversion in the opioid medication, we documented the recommended opioid via the PCCS. We obtained data from 74 patients, and the main opioids were hydromorphone (25 patients, 33.8% of the 74 patients), morphine (13 patients, 17.6%), tilidine (9 patients, 12.2%), oxycodone (8 patients, 10.8%) and fentanyl (8 patients, 10.8%). 

Pain relief 48 h after consultation is a hallmark of PC quality [12]. Hence, we evaluated all patients within the period from July 1, 2016 to May 32, 2018 (*n* = 1635), and whether they met this criterion. There were 372 patients who reported pain, out of which 282 suffered moderate to severe pain. Of the latter group, 254 (90%) patients described reduced pain levels, and 28 (10%) described the same or worse pain levels 48 h after consultation. 

##### Other Symptoms

We scheduled our inpatients’ premedication with respect to other symptoms. We detected that many of our patients took drugs for nausea (56.5%), excretion problems (45.5%), depression (36.5%), edemata (30.8%) and insomnia (28.8%). In addition, some patients took medication for neurological symptoms (16.3%) or dyspnea (11.6%). After our PC assessment, we recommended drug conversion because of excretion problems (19.8%), nausea (15.3%), insomnia (15.9%), dyspnea (6.5%) or depression (6.5%). In some cases, changes in drugs for the treatment of neurological issues (1.4%) were suggested.

We evaluated whether or not the physicians on the wards implemented our recommendations within 48 h. For the first 13 months, changes in the medication were implemented for excretion problems (49.5%), nausea (45.3%), insomnia (32.1%), dyspnea (46.9%), depression (59.4%) and neurological symptoms (57.1%). Moreover, we found that for male patients (41.2%), the recommended insomnia treatment was significantly more often realized when compared to female patients (14.8%, *p* < 0.01) (Table 2).

##### Other Care Services

We offered further care services besides our medical care. Many of our patients needed physiotherapy (81.2%), consultation for social issues (70%) and counseling about AHCD, healthcare proxies and/or further (outpatient) PC (55.9%). Furthermore, many patients received psychotherapy (43.3%) and spiritual care (36.9%). Other services that were offered included nutritional guidance (26.1%), dietary counseling (23.1%), music therapy (12.9%) and wound/stoma care (2.7%). 

## 4. Discussion

In the period from May 1, 2015 to May 31, 2018, the number of patients referred to the PCCS increased continuously. This trend has also been previously reported by other groups setting up a PCCS [21,22,23]. 

Furthermore, at the University Hospital in Munich, it was shown that only 30.3% of terminally ill cancer patients had been in contact with the PCCS before they died; in 54% of cases, the consultation was requested in the last week of the patient’s life [24]. This led to the question of why physicians still hesitate to request SPC. Interviews performed in a study were analyzed to facilitate the understanding of PCCS providers and requesting physicians [25]. The providers and requesting physicians agreed on what a PCCS should provide: handling and management of difficult symptoms, and support for complex psychosocial issues. Furthermore, access to the PCCS was relevant, including the possibility of communicating with the PC specialist in an informal way, and a fast response from the PCCS [25]. In our department, the PCCS was requested online, and within 24 h, the patient was consulted by a doctor or nurse, and responses to the requesting physician were given immediately. Furthermore, the PCCS was introduced officially by the head physician/nurses to every member of the different departments.

Another study reported the views of two different groups: the first group consisted of physicians with previous PCCS contact, and the second of those without any experience with PCCS [26]. Both groups agreed to manage many symptoms of patients with advanced disease, but at the same time, they felt less comfortable handling delirium, psychosocial issues and existential needs. As a result, 70% of the participants expressed that the most important barrier to PCCS access were the unrealistic expectations of patients and/or their families regarding prognosis [26]. In this study conducted at the Muenster University Hospital, the standard recommendations were implemented for most of the symptoms. Furthermore, both psychological and social issues were addressed by the PC team. To increase the prognostic awareness, diagnostic findings were discussed with the patient together with the physician in charge. The importance of this was underlined in a systematic review, which found that less than half of advanced cancer patients were aware of their prognosis [27]. 

Many oncologists tended to request SPC in cases of advanced disease, when the control of symptoms could no longer be managed [28]. This finding is explained as “persisting definitional issues” regarding SPC [28]. Further, a lack of information regarding end-of-life care and reasons for referral to PC, patient/family resistance, cultural differences and the present lack of service were described as perceived barriers to PCCS [29]. Offering PC education (didactic presentation and educational visits) and pocket-sized cards with referral criteria to the inpatient oncology team could be a way to increase the number of PCCS consultations, and to change referral reasons [30]. To address the abovementioned obstacles, we developed a flyer and prepared a lecture about the duties and responsibilities of a PCCS. With dedicated outreach activities, information about PCCS and its care possibilities was spread within the whole hospital.

Moreover, to set up a PCCS, it is important to know which patients to expect, and to be aware of the required core competencies of the designated PCCS team. In a study by Dhillon et al. at the M.D. Anderson Cancer Center of the University of Texas, the most common diagnoses were of thoracic/head and neck, gastrointestinal, genitourinary and gynecologic cancer [31]. Those are similar to the ones in our study, but they were ranked differently. This can be explained by the fact that Dhillon et al. grouped thoracic with head and neck cancer, while we considered them as two different tumor locations. As a consequence, team members ought to know how to operate for the various cancer entities, especially gastrointestinal, gynecological, head and neck (thoracic) and genitourinary cancer.

Additionally, many of our patients suffered from fatigue (82.7%), depression (68.6%), loss of appetite (64.7%), constipation (62.7%), pain (56.7%) and insomnia (54.9%). It is an important core competence to know how to treat these common symptoms. Within our service, the head physician is a gastroenterologist and could easily address gastrointestinal symptoms. Strong cooperation with our department of psycho-oncology, physical therapy and anesthesiology was established from the beginning. In the data provided by Delgado-Guay et al., reduced wellbeing (93%), fatigue (92%), pain (91%), loss of appetite (84%), insomnia (76%) and anxiety (70%) were common symptoms [32]. To conclude, the presentation of symptoms was similar to what we observed in our study; however, the representation differed, which could be explained by the different patient populations (only advanced cancer patients in the data of Delgado-Guay et al. vs. our wider patient population, with metastatic and non-metastatic cancer or non-cancer diagnosis). Taking the diversity of symptoms and problems into consideration, PCCS should provide inter- and multidisciplinary support for patients. However, we demonstrated that the symptom burden varies depending on the departments you are cooperating with (Figure 3). Ostgathe et al. showed that the burden of symptoms differs between cancer and non-cancer patients [33]. Dyspnea, weakness and fatigue occur more frequently in patients without tumor disease, whereas nausea, vomiting and loss of appetite are more frequent in cancer patients [33]; however, in that study, the analysis of the symptoms of palliative care patients was sorted according to a dichotomy, lacking differentiation into various entities or medical fields for a large group of non-cancer patients. Our study could, by contrast, not only provide insight into the qualitative burdens of the symptoms of referred patients (Figure 2C), but also decipher the frequency of symptoms presented, concerning the respective referring department (Figure 3). To our best knowledge, this is the first study to provide a comprehensive overview enabling both PCCS and the respective departments to quantify, and possibly anticipate or even be prepared for, the distinct and most frequent symptoms of palliative care patients in their medical field. These data may help to realize the competencies that PCCS team members need. The depiction of the frequency of department consultations, and the abovementioned PCCS algorithm (Figure 1), may provide helpful tools for the existing PCCS, as well as those yet to be established at other institutions.

In our study, admissions to internal medicine, gynecology and radiotherapy departments accounted for the highest numbers of referred patients. The regular ward rounds were performed once a week in those departments. This underlines the importance of the regular ward rounds for establishing an awareness of PC. Internal marketing was recognized as an important factor for raising the awareness of departments with regard to getting in contact with PC services [34]. The three main barriers to oncologists using PC services can be described as (1) the idea that PC is incompatible with cancer therapy; (2) the perception that PC has to be provided by oncologists; and (3) insufficient information about local PC services [35]. Implementing regular ward rounds once a week with the PCCS team, as well as supporting internal marketing, can break down these barriers; therefore, another important core competence is the ability to explain the role and possibilities of PC in regular ward rounds. Further, contact with the staff of different departments can help increase PC service awareness. 

We were able to show that 90% of our patients reported pain reduction 48 h after the first PCCS consultation. These patient-dependent evaluations are important for analysis of the quality of care in PC situations [36]. Ciemins et al. reported an improvement (reduction) in pain levels of 86% at discharge in a cohort of 282 patients [37]. Further, Hanks et al. demonstrated a significant improvement with respect to the most unbearable symptoms in the first week after consultation through the PCCS [38]. 

At the beginning of the PCCS, however, we could see that our recommendations for drug conversion were not always directly realized. There might be different reasons for this gap between recommendation and realization, e.g., the electronic recommendation was not seen by the physician in charge of the patient, or he/she has a different view concerning the symptoms or the need for PC. We recommend adopting an active approach, in which the implementation of the recommendation is evaluated the following day. 

In conclusion, palliative care aims at a better quality of life and control of symptoms for patients with life-limiting diseases. To implement a PCCS, you need to establish a multidisciplinary team [9,39,40]. with nurses trained in PC and physicians from different specialties, such as internal medicine, anesthesiology, neurology, oncology and physical therapy, as well as trained psychologists. Additionally, regular personal contact with the requesting physicians and nurses concerning patients from different departments can enhance the trust in and meaning of PCCS. Education is the key to creating an awareness of PC in the hospital, and to enabling access for patients.

## Figures and Tables

**Figure 1 ijerph-17-04977-f001:**
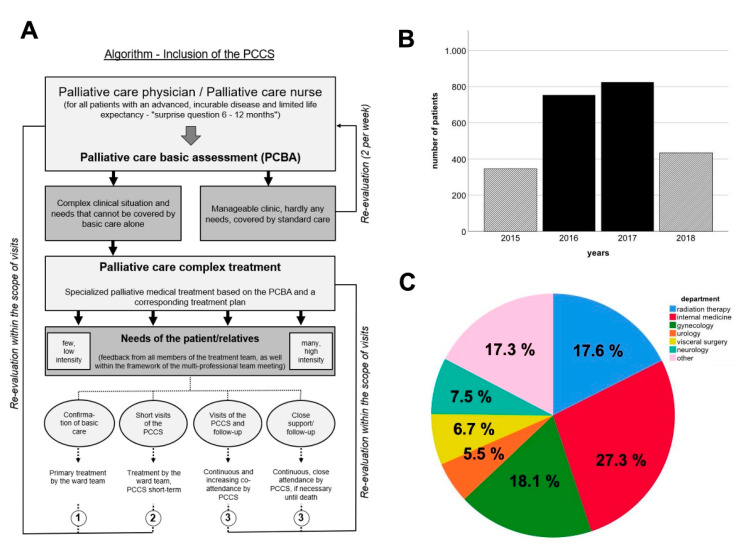
Algorithm of palliative care consultation service (PCCS) inclusion and analysis of patient number development as well as departmental allocation. Algorithm detailing how to match PCCS and a patient‘s situative need for palliative care (**A**). Patient number dynamics over the entire period of study (**B**). Pie chart comparing the amounts of patients referred to the PCCS by respective departments (**C**). “Other” includes nuclear medicine, cardiac surgery, neurosurgery, ophthalmology, vascular surgery, orthopedics, psychiatry, trauma surgery, dermatology, head and neck surgery, anesthesiology and ear, nose and throat medicine.

**Figure 2 ijerph-17-04977-f002:**
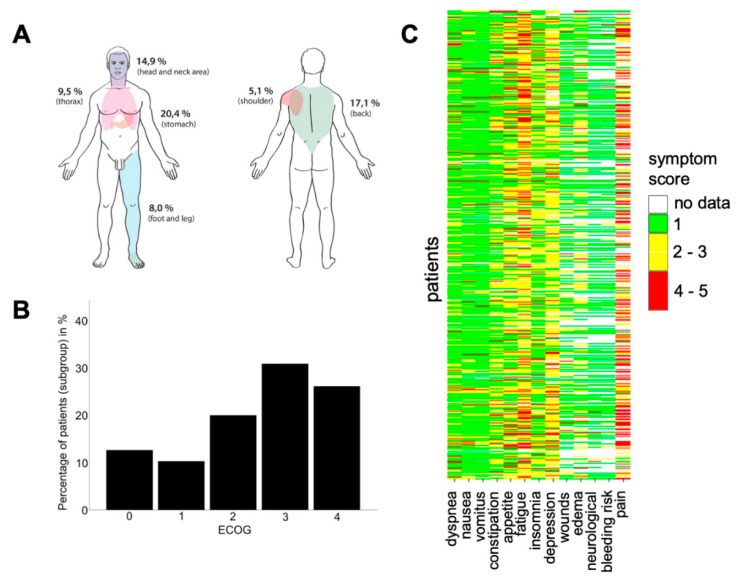
Evaluation of pain location, the burden of presented symptoms, and the ECOG performance status of referred patients at the first consultation with PCCS. Schematic overview of frequencies for the different pain locations of referred patients (**A**). ECOG status (0–4) of referred patients (**B**). Heatmap showing the burden of symptoms presented at the first PCCS consultation on a scale of stable (1), slightly reduced (2), reduced (3), greatly reduced (4) and unbearable (5) (**C**).

**Figure 3 ijerph-17-04977-f003:**
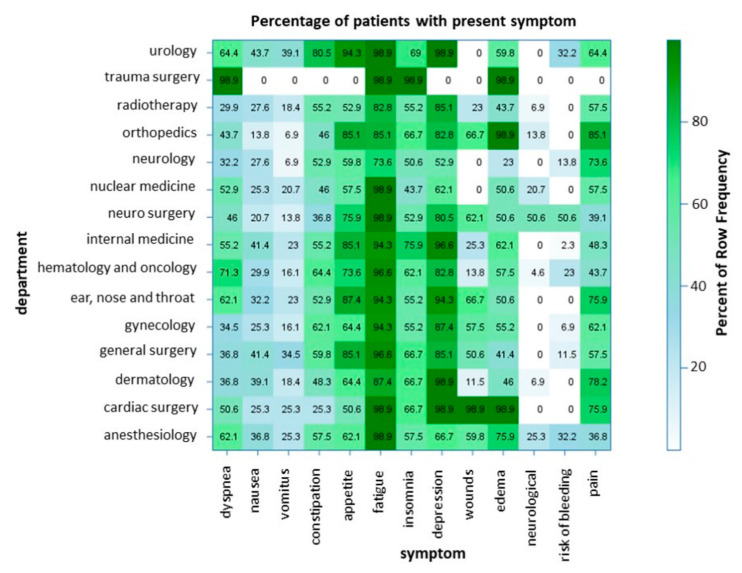
Correlation of the frequency of symptoms presented by patients referred to the PCCS and the respective departments.

**Table 1 ijerph-17-04977-t001:** Inpatient characteristics, main diagnoses, and Eastern Cooperative Oncology Group (ECOG) performance status of our subgroup analysis (for the first 13 months) (*n* = 490).

Median Age in Years (Range)	62 (17–95)
**Sex**	
Male	272 (55%)
Female	218 (45%)
**Social Status**	
Married (and living with wife/husband)	273 (55.7%)
Living alone	95 (19.4%)
Living with family	57 (11.6%)
Living with life partner	34 (6.9%)
Other	15 (3.1%)
No declaration	15 (3.1%)
**Advanced Healthcare Directive (AHCD)**	
Yes	215 (43.9%)
No	212 (43.3%)
No declaration	62 (12.7%)
**Healthcare Proxy**	
Yes	226 (46.1%)
No	200 (40.8%)
No declaration	63 (12.9%)
**Main Diagnoses**	
Cancer	429 (88%)
No cancer	61 (12%)

**Table 2 ijerph-17-04977-t002:** Analysis of recommendations and the realized implementation of drug conversions for inpatients (subgroup analysis, *n* = 490 *). Comparison of realization of recommendations between males and females for the respective symptom expressed by *p*-value of χ^2^ test.

Symptom	Total			Female			Male			*p*-Value
	Realized	Recommended	Proportion	Realized	Recommended	Proportion	Realized	Recommended	Proportion	
Pain	81	142	57.0%	39	66	59.1%	42	76	55.3%	0.422
Excretion problems	48	97	49.5%	18	40	40%	30	57	52.6%	0.434
Insomnia	25	78	32.1%	4	27	14.8%	21	51	41.2%	0.007
Nausea	34	75	45.3%	17	38	44.7%	17	37	45.9%	0.660
Depression	19	32	59.4%	8	13	61.5%	11	19	57.9%	0.727
Dyspnea	15	32	46.9%	3	8	37.5%	12	24	50.0%	0.112
Neurologic symptoms	4	7	57.1%	3	3	100%	1	4	25.0%	0.350

* Patients with and without recommended drug conversion.

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
