# Peer review of "Establishment of a Palliative Care Consultation Service (PCCS) in an Acute Hospital Setting"

_ijerph, 2020, doi:10.3390/ijerph17144977_

Round 1

Reviewer 1 Report

It is not clear what this phrase means There are only 15 PC and 17 hospice beds per million inhabitants. Does it mean 15 PC plus 17 hospice?

Section 3.2 also compare the median time to referral.

Table 1 - perhaps best to do a separate column for males and females and possibly note if males or females tended to receive a higher percentage of change of pain care.

This is purely optional, however, regarding end-of-life care I have been trying to get researchers to recognise that beds or PCS teams, etc per 1,000 population is an inappropriate measure and that figures should be quoted as beds, etc per 1,000 deaths.

I am also aware that there is additional research into symptom trends in the last years of life ...... which are somewhat like those in this study.

Author Response

Comment #1: It is not clear what this phrase means There are only 15 PC and 17 hospice beds per million inhabitants. Does it mean 15 PC plus 17 hospice?

Answer: We thank the reviewer for bringing up this important question. We changed the sentence to make the content clear (page 1, line 41). It means 15 PC plus 17 hospice beds.

Comment #2: Section 3.2 also compare the median time to referral.

Answer: We apologize for the inaccuracy, the mentioned referral times are the median referral times, we changed it accordingly in the manuscript (page 4, line 139).

Comment #3: Table 1 - perhaps best to do a separate column for males and females and possibly note if males or females tended to receive a higher percentage of change of pain care.

Answer: Thank you for this good proposal. We added this information to Table 2 (page 7, lines 240-243). The result is that females (59.1 %) exhibited a slightly more frequent though non-significant receipt of recommended pain care compared to males (55.3 %) (page 6, lines 211-212). Furthermore, we found that for male patients (41.2 %) recommended insomnia treatment was significantly more often realized compared to female patients (14.8 %, P < 0.01) (page 7, lines 237-239).

Comment #4: This is purely optional, however, regarding end-of-life care I have been trying to get researchers to recognise that beds or PCS teams, etc per 1,000 population is an inappropriate measure and that figures should be quoted as beds, etc per 1,000 deaths.

Answer: We totally agree with your concern. As stated in our study, recognition of PCCS should not be dependent upon the population in general. However, it would be very interesting to find a correlating parameter this recognition could be based upon. Unfortunately, the number of PCCS-Teams in Germany is inconsistent and unavailable for us at the moment. 

Comment #5: I am also aware that there is additional research into symptom trends in the last years of life ...... which are somewhat like those in this study.

Answer: Thank you for your comment. We definitely agree to the preexistence of symptom analyses in the literature. However, often cancer patients are analysed and for example compared to non-cancer cohorts. However, comprehensive analysis was lacking to our knowledge regarding not only symptom burden and presence in different disease entities, but rather – and this is what we also considered very valuable when starting this study for ourself future mode of operation – our study provides insight into the presence of symptoms in the different departments of our clinic. This is important not only for the set up of our/a PCCS team and its qualification, but also departments should be best prepared for the respective symptoms most prevalent in their clinic (of which we seek to provide this overview). We added a description of that on page 9, lines 312-320.

Reviewer 2 Report

A very good article that, on the one hand, describes the barriers and obstacles
in the implementation of the PCCS in an acute hospital and, on the other, helps
to highlight its outcomes. This sharing is important for the empowerment of
palliative care.
On page 78 you wrote "A SPC physician or nurse ..." is the first time that this
acronym appears, it was important to write its meaning.
On page 92 you claim that you have 1/2 psycho-oncologist and 1/2 social worker,
I think it would be better to write half-time weekly or monthly as appropriate.
Finally, you state that patients are referred to by different doctors when they
identify palliative care needs. I did not understand from what I read if the
nursing team in any way references patients

Author Response

A very good article that, on the one hand, describes the barriers and obstacles in the implementation of the PCCS in an acute hospital and, on the other, helps to highlight its outcomes. This sharing is important for the empowerment of palliative care.

Comment #1: On page 78 you wrote "A SPC physician or nurse ..." is the first time that this

acronym appears, it was important to write its meaning.

Answer: Thank you very much for your helpful comment. We added the meaning of the acronym (page 3, line 97). 

Comment #2: On page 92 you claim that you have 1/2 psycho-oncologist and 1/2 social worker,

I think it would be better to write half-time weekly or monthly as appropriate.

Answer: We completely agree with you, it is not written comprehensible. We corrected it in the text (page 3, line 111).

Comment #3: Finally, you state that patients are referred to by different doctors when they

identify palliative care needs. I did not understand from what I read if the nursing team in any way references patients.

Answer: Thank you for this important notice. In our German health care system a doctor refers patients to other departments. However, in clinical practice the nurses and the doctors discuss the medical needs of their patients together. In the text we added that the different doctors decide “with their team” if there is a palliative care need (page 3, line 97). Formally, the referral has to be performed by a doctor in our health care system.

Reviewer 3 Report

Thank you for an interesting submission. It is yet promising to read that palliative care consultation is continuously under demand, where necessary. The paper is scientifically sound and clearly presents the results. Yet, there are areas that require some improvement.

The introductory section flows well but it does not take into account the myriad of work undertaken in this area, not purely from a medical or clinical standpoint. I wonder if the authors are familiar with the work of Brinkman-Stoppelenburg, for example. I would also encourage the authors to consider bibliography outside of clinical practice; there is abundant work done.

Methodologically, the section can be improved with some information about ethics and consent; explicit guidance on how the data was collected, and perhaps some clarity about the inclusion of patients; in the introduction there is a mention about life-threatening disease, but in the methodology the authors refer to life-limiting. Further, can you clarify what you mean by 1/2 psychologist and 1/2 social worker that partook in the study? I am confused about them being half each; perhaps it is not well described in the text?

The conclusions need some addition to highlight the original contribution of this study. The contribution needs to original.

Hopefully these comments help improve the text.

Author Response

Thank you for an interesting submission. It is yet promising to read that palliative care consultation is continuously under demand, where necessary. The paper is scientifically sound and clearly presents the results. Yet, there are areas that require some improvement.

Comment #1: The introductory section flows well but it does not take into account the myriad of work undertaken in this area, not purely from a medical or clinical standpoint. I wonder if the authors are familiar with the work of Brinkman-Stoppelenburg, for example. I would also encourage the authors to consider bibliography outside of clinical practice; there is abundant work done.

Answer: Thank you very much for these comments. We agree with you and found out about the interesting work of Brinkman-Stoppelenburg and their important results about quality of dying and the multidimensional concept behind it (page 2, line 58-60). We also expanded the text with information about the important role of the different perceptions of patients, nurses, doctors and relatives (page 2, line 70-71). Finally, we highlighted the most important needs in the end-of-life care from the patients’ and family members’ perspective (page 2, lines 72-74).

Comment #2: Methodologically, the section can be improved with some information about ethics and consent; explicit guidance on how the data was collected, and perhaps some clarity about the inclusion of patients; in the introduction there is a mention about life-threatening disease, but in the methodology the authors refer to life-limiting.

Answer: As stated in the methods section, our study design was retrospective. With admission to the hospital all patients agreed that their medical history was stored in the clinical system. After permission of the local ethic committee, we analyzed pseudonymous data. In addition, you are completely right, we were imprecise in the introduction. We corrected it to “life-limiting disease” (page 1, line 34).

Comment #3: Further, can you clarify what you mean by 1/2 psychologist and 1/2 social worker that partook in the study? I am confused about them being half each; perhaps it is not well described in the text?

Answer: We totally agree with you. We wanted to express, that both the psycho-oncologist and the social worker work for our team half-time weekly. We corrected it in the text (page 3, line 111).

Comment #4: The conclusions need some addition to highlight the original contribution of this study. The contribution needs to original. Hopefully these comments help improve the text.

Answer: Thank you for this important comment. We sought to shed light on the relevance and originality of our study in discussion section (page 9, lines 312-322).